# Comparative Efficacy and Safety of Statin Monotherapy and Statin plus Ezetimibe Combination in a Real-World Setting

**DOI:** 10.3390/diseases11040168

**Published:** 2023-11-13

**Authors:** Marat V. Ezhov, Igor V. Sergienko, Sergey M. Kryzhanovskiy, Kirill S. Manko, Elena V. Timoshina

**Affiliations:** 1Federal State Budgetary Institution, National Medical Research Center of Cardiology Named after Academician E.I. Chazov of the Ministry of Health of the Russian Federation, 121552 Moscow, Russia; igorcardio@mail.ru; 2CRO Ligand-Research LLC, 119048 Moscow, Russia; sergey.kryzhanovskiy@ligand-research.com; 3JSC AKRIKHIN, 115054 Moscow, Russia; k.manko@akrikhin.ru (K.S.M.); e.timoshina@akrikhin.ru (E.V.T.)

**Keywords:** hypercholesterolemia, statins, ezetimibe, lipid-lowering therapy, effectiveness, safety

## Abstract

Background: The objective of this study was to conduct a comparative evaluation of the effectiveness of ezetimibe in combination with statins or statin monotherapy in patients with hypercholesterolemia in a real-world setting. Methods: It was a retrospective multicenter observational study conducted in Russia. We included patients who received statins or a combination of statins with ezetimibe for ≥3 months. The primary endpoint of this study was the frequency of achieving low-density lipoprotein cholesterol (LDL-C) goal levels at the time of enrollment in the study (%). Results: The full analysis set consisted of 1000 patients: 250 subjects in the statin monotherapy group and 750 subjects in the combination group. The groups did not differ in clinical, demographic, or laboratory variables, except for a higher prevalence of hypertension and higher baseline lipid values in the statin monotherapy group. During treatment, the LDL-C concentration decreased by 1.10 ± 1.04 mmol/L (change of −27.5 ± 28.5% from baseline) in the statin monotherapy group and by 1.55 ± 1.17 mmol/L (change of −38.2 ± 25.6% from baseline) in the combination therapy group, *p* < 0.001. The target LDL-C level was achieved in 22.4% of the patients in the monotherapy group compared with 28.8% of the patients in the combination therapy group, *p* = 0.049. Conclusions: In real-world clinical practice, statin/ezetimibe combination therapy demonstrated a more frequent achievement of target LDL-C levels compared with statin monotherapy. The addition of ezetimibe to statin therapy increased the probability of achieving LDL-C level goals by 29%.

## 1. Introduction

Cardiovascular disease (CVD) remains one of the leading causes of death in many countries. Hypercholesterolemia is the main causal atherosclerosis risk factor that is poorly controlled worldwide [1]. The first and main option for low-density lipoprotein cholesterol (LDL-C) control is a statin at a maximally tolerated dose. However, numerous studies demonstrated the low frequency of LDL-C goal attainment with statin monotherapy for high and very high cardiovascular risk persons. The use of ezetimibe in combination with a statin is a more effective choice for a significant decrease in LDL-C concentration [2]. One large randomized clinical trial demonstrated a significant cardiovascular outcomes reduction with statin plus ezetimibe compared to statin-alone treatment in patients after acute coronary syndrome [3]. These data allowed for providing the IA class and level of evidence for ezetimibe treatment for LDL-C goal achievement [2]. Despite these guidelines, recent large cohort studies showed the low frequency of combination therapy in real practice in European countries [4,5]. The latest paradigm is to start treatment of very high risk subjects with a combination of statin with ezetimibe [6].

Russia is on the list of very high cardiovascular risk countries due to a high population level of cardiovascular mortality [7]. One of the reasons for the poor statistics is the high prevalence of lipid disorders among adults [8]. Moreover, most people are unaware about their health status and risk factors profile, including the presence of elevated total cholesterol [9]. Poor adherence to statins is a worldwide problem and another reason for the non-achievement of LDL-C goals. Ezetimibe combined with any statin should be considered as an essential step for the better treatment of patients with a high probability of cardiovascular events development [10]. A large simulation study of six countries including Russia provided confirmation that both the free- and fixed-dose statin plus ezetimibe combination will substantially allow for the prevention of major adverse cardiovascular events [11].

The results obtained in randomized clinical trials may differ from real-world clinical practice. In this regard, post-marketing clinical studies are necessary to obtain the efficacy and safety of lipid-lowering therapy in the real world. Any new data obtained from different countries should enlarge the body of evidence for the best clinical practice.

The objective of this study was a comparative evaluation of the effectiveness and safety of ezetimibe in combination with statins or a statin monotherapy in real clinical settings in Russia.

## 2. Material and Methods

### 2.1. Study Setting

This study was conducted at clinical sites involved in the routine management of patients with dyslipidemia. After a preliminary assessment of 109 clinical sites, 48 centers in various regions of the Russian Federation agreed to participate in this study.

### 2.2. Participants

During the period from 29 June 2021 to 25 November 2021, all the study sites included 1000 patients. The inclusion criteria were as follows: 1. Age > 18 years. 2. Lipid-lowering therapy with a statin or statin in combination with ezetimibe in a stable dosing regimen for three or more months and no more than two years before the study enrolment. 4. A willingness and ability to sign an informed consent form to participate in the study. 5. The availability of the primary medical documentation, which allows for an assessment of all the parameters necessary for this study from the moment of initiating lipid-lowering treatment.

This study did not include patients with established or suspected familial hypercholesterolemia, intake of fibrates and omega-3 polyunsaturated fatty acids, and treatment with methods of extracorporeal filtration and/or plasmapheresis. Clinically significant hepatic and/or renal impairment, hypothyroidism, ezetimibe monotherapy, and statin intolerance at any dose were also exclusion criteria. All drugs were prescribed in accordance with the recommendations given by the attending physician as a part of routine clinical practice and in accordance with the routine practice of the study site. All therapy was administered prior to data collection and prior to enrollment in this study. There were no study dropouts due to the retrospective design. All the patients signed an informed consent form. All the patients completed this study.

### 2.3. Study Design

The UNISON study is a retrospective observational study of the efficacy and safety of statin monotherapy or a statin in combination with ezetimibe in patients with hypercholesterolemia. Considering a retrospective observational study design, which meant only the collection of data contained in medical documentation, no specific procedures for the patients were conducted, except for signing the informed consent form. Before signing the informed consent form, the study physician evaluated the criteria eligibility for this study. The information from the outpatient medical record was evaluated from the initiation of lipid-lowering therapy (3–24 months prior to the enrollment) up to the date of enrollment (Visit 1). The data collected from the medical records were recorded in the electronic case report form (eCRF) by the investigators.

### 2.4. Ethical Approval

This study was organized in accordance with the principles of good clinical practice, the Declaration of Helsinki, and applicable to local regulations. This study was initiated and conducted by the Russian National Atherosclerosis Society (RNAS) and the Contract Research Organization Ligand Research under the sponsorship of JSC AKRIKHIN. Every patient was informed about all aspects of this study and had an opportunity to have any study-related question answered before signing the informed consent form. This study was approved by the Independent Interdisciplinary Committee for ethical expertise of clinical trials (Moscow, Russia), protocol #7 dated 23 April 2021. This study was registered at ClinicalTrials.gov (NCT04895098).

### 2.5. Study Procedures

The assessments for the patients provided routine demographic data, including age, sex, height, body weight, waist circumference, family history of CVD, personal CVD and its duration, prior revascularization procedures, concomitant diseases, and current therapy, including lipid-lowering drugs. The following parameters of the lipid profile were analyzed at baseline and over time: total cholesterol, TG, HDL-C, and LDL-C. Also, at baseline and over time, the safety of lipid-lowering therapy was determined based on the levels of transaminases (aspartate aminotransferase (AST) and alanine aminotransferase (ALT)), as well as the level of creatine kinase (CK).

Information about adverse drug reactions (ADRs) was collected as a part of the routine practice of the doctor. For the safety analysis, the data specified in the outpatient records or other patient records were used. ADRs were coded according to the Medical Dictionary for Regulatory Activities (MedDRA).

### 2.6. Endpoints

The primary endpoint of this study was the frequency of achieving the target levels of LDL-C (in accordance with the recent European dyslipidemia guidelines) at the time of enrollment in the study (%) [2]. The secondary endpoints included the mean change in LDL-C from the start of statin therapy to the study enrollment (absolute difference and percent change from baseline); the mean change in the total cholesterol level from the start of statin therapy to enrollment in the study (percent change and absolute difference); the mean change in the HDL-C level from the start of statin therapy to enrollment in the study (percent change and absolute difference); and the mean change in the TG level from the start of statin therapy to enrollment in the study (percent change and absolute difference).

### 2.7. Statistical Methods

The data of all the patients included in this study were entered into an electronic database via an electronic statistical analysis, which was carried out using R version 4.1.2 (The R Foundation for Statistical Computing, Vienna, Austria). Descriptive statistics were used, and the following values were calculated: the arithmetic mean, 95% confidence interval (CI) for the mean (unless otherwise indicated), standard deviation, median, interquartile range, and minimum and maximum for continuous data. The nominal/discrete data were processed by calculating the proportion of the absolute number of observations. The Shapiro–Wilk test was used to assess the normality of the distribution of the quantitative data. The qualitative data were presented as the absolute frequencies (number of observations), proportions (percentages), and 95% confidence interval (CI) (unless otherwise indicated). The discrete data were compared across the treatment groups using the chi-squared test/Fisher exact test, and the continuous data were compared using the unpaired Student *t*-test for normally distributed data or the non-parametric Mann–Whitney test for non-normally distributed data. Comparisons with set levels were performed using the Wilcoxon test. All the tests were two-sided and performed at a 5% level of significance. In general, the substitution of missing data was not performed, i.e., missing data were not replaced by the calculation or creation of new data but were handled as ‘missing’ in the statistical evaluation.

## 3. Results

### 3.1. Study Population

The full analysis set consisted of 1000 patients (100%): 250 participants in the statin monotherapy group and 750 in the ezetimibe plus statin therapy group. Due to the retrospective design, there were no patient dropouts and protocol deviations. The groups did not differ in clinical, demographic, or laboratory variables, except for a higher prevalence of hypertension and higher baseline lipid values in the statin monotherapy group. (Table 1).

At the enrollment, most patients were receiving atorvastatin (54.2%) and rosuvastatin (41.1%) (Figure 1).

### 3.2. Efficacy of Treatment, FAS (Full Analysis Set) Population

#### 3.2.1. Achievement of the LDL-C Target

The primary efficacy endpoint was the frequency of achieving the target LDL-C levels (according to the 2019 ESC/EAS Guidelines for the management of dyslipidemia as well as local guidelines) at the time of study enrollment. The LDL-C target was achieved in 22.4% (56/250) of the patients [95% CI 17.4–28.1] in the monotherapy group versus 28.8% of the patients (216/750) [95% CI 25.6–32.2] in the combination therapy group (Figure 2). The differences between the groups in the frequency of achieving the LDL-C target were statistically significant: OR 0.714 (0.499–1.009), *p* = 0.049.

#### 3.2.2. LDL-C

During treatment, the LDL-C concentration decreased by 1.10 ± 1.04 mmol/L (change of −27.5 ± 28.5% from baseline) in the statin monotherapy group and by 1.55 ± 1.17 mmol/L (change of −38.2 ± 25.6% from baseline) in the combination therapy group, *p* < 0.001. 

#### 3.2.3. Total Cholesterol

The total cholesterol reduction was 1.25 ± 1.12 mmol/L (change of −20.4 ± 19.0% from baseline) in the statin monotherapy group and 1.76 ± 1.27 mmol/L (change of −27.7 ± 18.0% from baseline) in the combination therapy group, *p* < 0.001.

#### 3.2.4. HDL Cholesterol

During therapy, there was a slight increase in HDL-C: by 0.020 ± 0.376 mmol/L in the statin monotherapy group (+5.0 ± 33.1% from baseline) and by 0.036 ± 0.397 mmol/L (+8.4 ± 38.5% from baseline) in the combination therapy group, *p* < 0.001.

#### 3.2.5. Triglycerides

The TG level decreased from baseline in both groups. The change was −0.195 ± 0.907 mmol/L (−0.1 ± 61.0% from baseline) in the statin monotherapy group and −0.424 ± 1.559 mmol/L (−13.9 ± 44.6% from baseline) in the statin plus ezetimibe combination therapy group (*p* < 0.001) (Figure 3 and Figure 4).

### 3.3. Target Achievement in CV Risk Differentiated Subgroups

An analysis of the subgroups formed based on the level of cardiovascular risk and the intensity of statin therapy revealed that the LDL-C target achievement proportion in patients at high cardiovascular risk, who received high-intensity statin therapy, was higher in the subgroup of patients treated with ezetimibe and statins (32% vs. 25% in the statin monotherapy group, *p* = 0.492), while these proportions in the subgroup of patients at very high risk were 24% and 16%, respectively (*p* = 0.059). Also, the target levels were more frequently achieved in the subgroup of patients at very high risk, who received moderate-intensity statin therapy: 33% in the ezetimibe plus statin group vs. 15% in the statin monotherapy group (*p* = 0.035). This finding was not confirmed in the other subgroups due to the small sample sizes (Table 2).

### 3.4. Safety

During the study, 17 adverse events (AEs) were reported in 16/1000 (1.6%) patients. Of these, two (33%) events in the monotherapy group and six (55%) events in the combination therapy group were mild. Three (50%) AEs in the former group and four (36%) AEs in the latter group were moderate. One severe AE was observed in each of the groups: one (17%) and one (9%), respectively.

According to the protocol, AEs of special interest included AEs associated with hepatic impairment or muscle tissue and major cardiovascular events. There were no statistically significant differences between the groups in the frequency of musculoskeletal and connective tissue disorders (zero (0%) and three (0.4%), respectively), hepatobiliary disorders (one (0.4%) and two (0.3%), respectively), the need for revascularization procedures (two (0.8%) and three (0.4%), respectively), cardiac events (unstable angina) (one (0.4%) and one (0.1%), respectively), as well as ischemic stroke (one (0.4%) and zero (0%), respectively).

## 4. Discussion

This is the first large-scale study of the use of ezetimibe in real-world clinical practice in Russia aimed to analyze the achievement of the new LDL-C targets specified in the latest clinical guidelines. This multicenter, retrospective, comparative, observational study of the efficacy and safety of statin monotherapy and the combination of statins with ezetimibe demonstrated that the LDL-C target achievement rate in the statin monotherapy group was only 22.4%. Combination therapy with statins and ezetimibe is associated with a more frequent achievement of LDL-C targets with good tolerability and no increase in the incidence of adverse events. Our data are very similar to the European DA VINCI study that showed that high-intensity statin monotherapy in very high risk primary and secondary prevention patients allowed for achieving LDL-C targets in 17% and 22%, respectively [4].

Several real-world studies in different countries demonstrated the comparative advantage of ezetimibe added to statins to reach LDL-C target levels [12,13,14,15]. In a Chinese retrospective cohort study of 1727 ASCVD patients, the addition of ezetimibe to a statin produced the achievement rates of LDL-C below 1.8 and 1.4 mmol/L over the first year, as high as 50.6 and 25.6%, respectively. However, in the second and third follow-up years, these rates decreased to 31.3, 30.3% and 15.5, 16.5%, respectively. The multivariable analysis showed that male sex, the combined use of atorvastatin or rosuvastatin with ezetimibe, better adherence, and smoking cessation were associated with a higher achievement rate [12]. A Spanish retrospective, observational study included 1570 ASCVD patients. They were treated with ezetimibe combined with atorvastatin (47.8%) or rosuvastatin (52.2%) in a high-intensity regimen. Despite these combinations, LDL-C below 1.4 mmol/L was not reached in about 70% of the participants [13]. The Korean retrospective study analyzed electronic medical records from 4252 patients treated between 2015 and 2017 in two tertiary care medical centers. Only those who switched to the statin/ezetimibe combination after statin monotherapy were enrolled. Enhancing the lipid-lowering therapy provided additional significant LDL-C level reduction by 31–41% depending on statin intensity and the achievement of the LDL-C levels < 1.8 mmol/L in about 90% of the subjects. A subgroup analysis was performed of the better efficacy of rosuvastatin/ezetimibe than the atorvastatin/ezetimibe combination within the same statin intensity [14]. A retrospective analysis of the electronic medical records of 311,242 very high risk outpatients showed that the addition of ezetimibe in patients already prescribed a statin reduced LDL-C by an additional 24%, with a greater reduction by 28% with a fixed dose compared to a free combination (19.4%; *p* < 0.0001) with the achievement of an LDL-C level of <1.8 mmol/L in 31% and 21% of the cases, respectively [15]. Even the latest data from an observational, prospective SANTORINI study of around 9000 patients at high or very high risk between 2020 and 2021 in 14 European countries documented that the landscape of lipid-lowering therapy remained unchanged [5]. Considering that 24% were on combination therapy, 54% were receiving statins monotherapy, and 22% were not treated at all, the current European LDL-C goals were achieved only in each fifth subject [5].

The efficacy in terms of lipid metabolism, tolerability, and the safety of ezetimibe, both in monotherapy and in combination with statins, including the fixed-dose combination of simvastatin or rosuvastatin or atorvastatin at different doses, was extensively evaluated in randomized clinical studies. Combination therapy with ezetimibe and atorvastatin at a dose of 10 mg was associated with a 53% reduction in LDL-C (comparable with the effect of atorvastatin monotherapy at a dose of 80 mg, 54%) [16]. The addition of ezetimibe to rosuvastatin 40 mg resulted in a 70% reduction in LDL-C (compared to a 57% reduction in LDL-C in the rosuvastatin 40 mg group) [10]. The addition of ezetimibe to ongoing statin therapy for 6 weeks resulted in an additional 25% decrease in LDL-C, while the decrease in patients who received a placebo and continued statin therapy was only 3.7% [17].

The data from an 8-week, double-blind, multicenter, randomized, controlled phase III study (I-ROSETTE) showed that the average change in the level of LDL-C in all groups of combination therapy with rosuvastatin plus ezetimibe was more than 50%. The number of patients who had achieved LDL-C targets at week 8 was significantly greater in the ezetimibe plus rosuvastatin group (180 (92.3%) of 195 patients) than in the rosuvastatin monotherapy group (155 (79.9%) of 194 patients) (*p* < 0.001) [18]. Thus, combination therapy with statins and ezetimibe provides an additional 18–25% reduction in LDL-C and significantly increases the number of patients achieving the LDL-C target.

In contrast to the fixed-dose combinations of statins and ezetimibe, ezetimibe has an advantage of its possible addition to any statin at any dose. In our study, the patients treated with combination therapy were significantly more likely to achieve the target lipid levels, 28.8% vs. 22.4% in the monotherapy group, and the LDL-C concentration decreased more significantly in the combination therapy group.

The results of the UNISON study add new information to the available evidence from randomized studies and real-world settings and confirmed a rather low LDL-C target achievement rate in patients with high and very high cardiovascular risk. The study demonstrated superior efficacy of combination therapy with statins and ezetimibe compared with statin monotherapy in terms of more patients achieving target levels of LDL-C. These conclusions are supported by an additional analysis, which demonstrated that the addition of ezetimibe to both high-intensity statin therapy and moderate-intensity statin therapy led to statistically significant results in achieving lipid targets, changes over time, and a percent reduction in LDL-C. Combination therapy was more effective when used in patients with very high cardiovascular risk. It should also be mentioned that the addition of ezetimibe to the therapy did not increase the risk of adverse reactions (the proportion of adverse reactions was the same in both groups). Meanwhile, our study provides important and additional information received from almost 50 clinical centers in Russia and from thousands of patients that statins with ezetimibe represent the optimal paradigm for improving the treatment of LDL-C.

## 5. Study Limitations

Our study has several limitations. First, it has a retrospective design with the inclusion of patients receiving statin monotherapy and a combination of various statins with ezetimibe in a ratio of 1:3, with a total number of 1000 treated patients. We have data from a randomized controlled trial, according to which the addition of ezetimibe to a statin improves the clinical outcomes of patients with coronary heart disease due to better control of LDL-C levels [3]. However, there is not enough data from the real world. Most studies evaluating the benefits of ezetimibe in addition to statins have also been retrospective [12,13,14,15]. Insufficient use of ezetimibe in many countries depends more on the inertia of physicians and the resistance of patients to current methods of treatment of dyslipidemia. Second, because the retrospective data from patient records were analyzed, there was no control over the patients’ adherence to therapy. It was shown that in the case of the free combination, the chance of missing tablets of either the statin or ezetimibe is increased [15]. If some of our patients on combination therapy had low adherence to it, the real number of those on LDL-C targets could be higher. Third, several patients had non-optimal starting doses of statins that were not adjusted in a timely manner. But again, this fact demonstrates that in routine practice many physicians have concerns regarding maximal doses of statins. The DA VINCI study also provides evidence that high-intensity statins are used only 22% and 42% in primary secondary prevention groups, respectively [4]. Considering the similarity in the results of the DA VINCI and UNISON studies, we recognize that there is significant room for improvement in high and very high cardiovascular patients’ management both in Europe and in Russia. At the same time, because of the much higher cardiovascular morbidity and mortality in Russia, greater efforts should be provided by our healthcare system to increase both general practitioners’ and their patients’ awareness and education on the current opportunities of hypercholesterolemia management. If the free combination of a statin with ezetimibe will be used widely and in accordance with the current dyslipidemia guidelines, approximately 342,000 major adverse cardiovascular events may be prevented in Russia over 5 years compared with the continuation of clinical practice as of the beginning of 2020 [11].

Despite all the limitations, our study provides new and valuable information on the effectiveness and safety of statins in combination with ezetimibe therapy in a real-world setting.

## 6. Conclusions

In real-world clinical practice, statin/ezetimibe combination therapy demonstrates more frequent achievement of target LDL-C levels compared with statin monotherapy. The addition of ezetimibe to statin therapy increases the probability of achieving LDL-C goals by 29%.

## Figures and Tables

**Figure 1 diseases-11-00168-f001:**
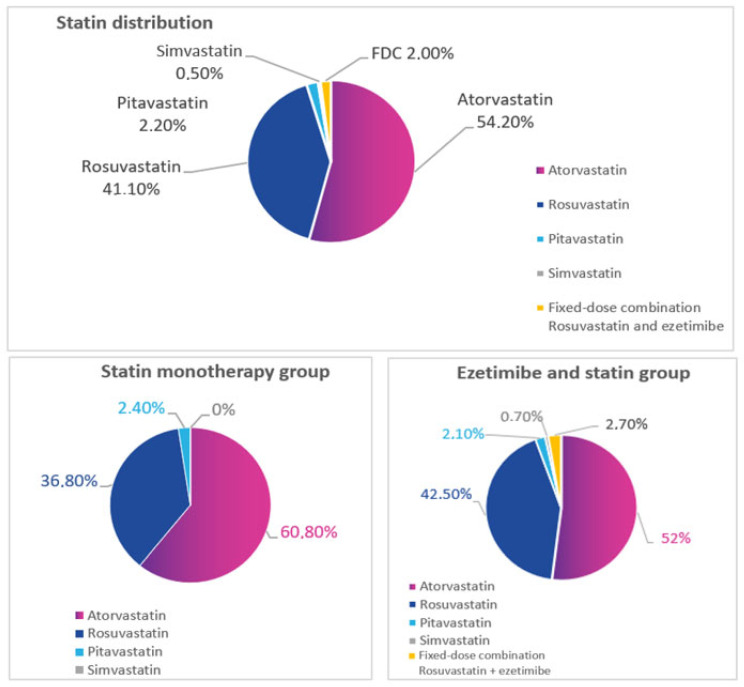
Treatment regimen in the study.

**Figure 2 diseases-11-00168-f002:**
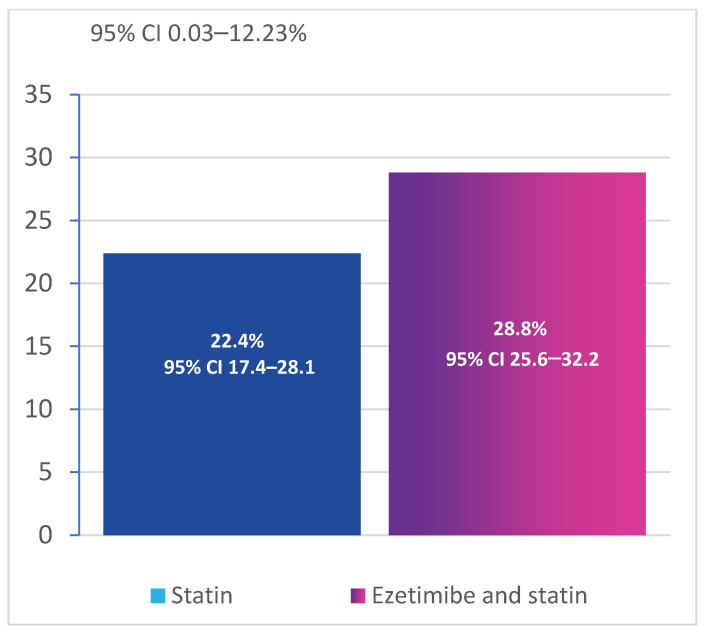
Frequency of LDL cholesterol (mmol/L) target achievement (CI, confidence interval).

**Figure 3 diseases-11-00168-f003:**
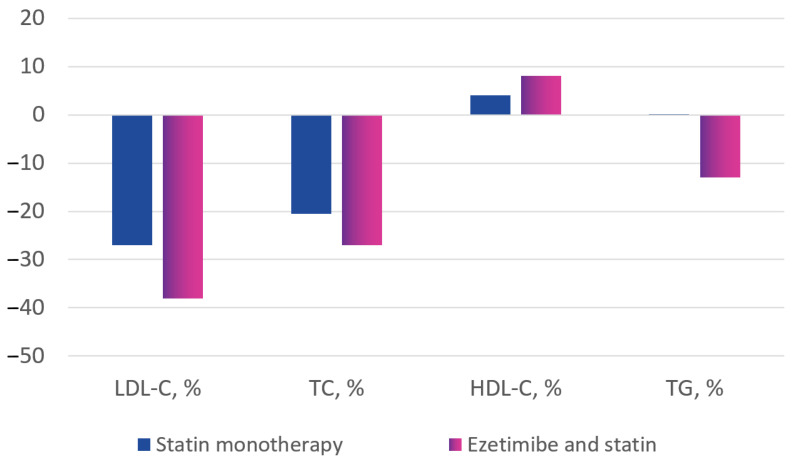
Lipid profile changes in the treatment groups, % change from baseline.

**Figure 4 diseases-11-00168-f004:**
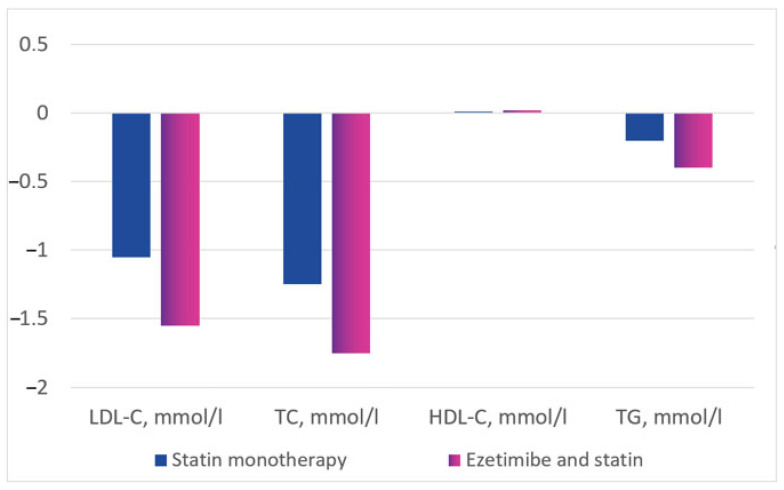
Lipid profile changes from baseline in the treatment groups, mmol/L.

**Table 1 diseases-11-00168-t001:** Baseline characteristics of study population (*n* = 1000).

Parameters	Statin Monotherapy (*n* = 250)	Statin Plus Ezetimibe (*n* = 750)	*p*
Males, *n* (%)	155 (62.0)	429 (52.7)	0.18
Age, years	59.9 ± 10.9	60.2 ± 10.5	0.55
Body weight, kg	84.4 ± 15.2	84.2 ± 15.5	0.75
Smoking status			
Former smoker, *n* (%)	46 (18.4)	143 (19.1)	0.49
Current smoker, *n* (%)	51 (20.4)	128 (17.1)	
Never smoked, *n* (%)	153 (61.2)	479 (63.9)	
Concomitant diseases			
Coronary heart disease	140 (56.0)	424 (56.5)	0.88
Acute coronary syndrome	7 (2.8)	38 (5.1)	0.19
Myocardial infarction	35 (14.0)	138 (18.4)	0.11
Hypertension	202 (80.8)	559 (74.5)	0.044
Type 2 diabetes mellitus	40 (16.0)	108 (14.4)	0.54
Ischemic stroke	7 (2.8)	28 (3.7)	0.62
Chronic kidney disease	40 (16.0)	152 (20.3)	0.14
Lipid parameters			
Total cholesterol	4.20 (3.53, 5.10)	3.90 (3.40, 4.59)	<0.001
Low-density lipoprotein cholesterol	2.20 (1.67, 3.00)	1.90 (1.50, 2.57)	<0.001
High-density lipoprotein cholesterol	1.20 (1.00, 1.41)	1.21 (1.02, 1.45)	0.094
Triglycerides	1.41 (1.03, 1.90)	1.30 (0.96, 1.83)	0.049
Revascularization surgery			
Coronary stent placement	80 (32.0)	270 (36.1)	0.25
Percutaneous coronary intervention	16 (6.4)	48 (6.4)	1.00
Coronary artery bypass grafting	10 (4.0)	41 (5.5)	0.36
Peripheral artery surgery	2 (0.8)	14 (1.9)	0.38
Concomitant therapy			
Beta-blockers	152 (60.8)	468 (62.4)	0.65
Angiotensin-converting enzyme inhibitors	93 (37.2)	295 (39.3)	0.55
Angiotensin II receptor antagonists	74 (29.6)	217 (28.9)	0.84
Calcium channel blockers	65 (26.0)	211 (28.1)	0.51
Diuretics	79 (31.6)	250 (33.3)	0.61

**Table 2 diseases-11-00168-t002:** LDL-cholesterol level goals achievement depending on risk category and treatment regimen.

Cardiovascular Risk Category	Statin Therapy Intensity	Statin Monotherapy		Ezetimibe and Statin		*p*	Δ (Difference between the Groups)
		*n*	%	*n*	%		
High risk	High	24	25	93	32	0.492	7%
	Moderate	42	33	110	26	0.394	−7%
Very high risk	High	103	16	353	24	0.059	8%
	Moderate	40	15	98	33	0.035	18%

## Data Availability

The data presented in this study are available on request from the corresponding author. The data are not publicly available due to national regulations.

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
