# Peer review of "Comparative Efficacy and Safety of Statin Monotherapy and Statin plus Ezetimibe Combination in a Real-World Setting"

_diseases, 2023, doi:10.3390/diseases11040168_

Round 1
Reviewer 1 Report (Previous Reviewer 1)
Comments and Suggestions for Authors
This version is good for me.
Author Response
Dear Reviewer,
We express our sincere gratitude for your help regarding the analysis and improving our paper.
Respectfully yours,
Marat Ezhov
Reviewer 2 Report (Previous Reviewer 2)
Comments and Suggestions for Authors
The authors respond to comments. However, only observational studies showed the LDL-C lowering effect of ezetimibe+statin and no data to adjust for confounding factors beyond RCTs. I regret to say that the substantial scientific value may not be worth publishing.
Comments on the Quality of English LanguageNo major concerns about English quality.
Author Response
Dear Reviewer, We express our sincere gratitude for your help regarding the analysis and improving our paper. We added one sentence at the end of the Discussion section and critically revised and expanded the Study limitations section, focusing readers' attention on the novelty of the study.
Respectfully yours,
Marat Ezhov
Reviewer 3 Report (Previous Reviewer 4)
Comments and Suggestions for Authors
Much improved of the revised.
Author Response
Dear Reviewer,
We express our sincere gratitude for your help regarding the analysis and improving our paper.
Respectfully yours,
Marat Ezhov
This manuscript is a resubmission of an earlier submission. The following is a list of the peer review reports and author responses from that submission.
Round 1
Reviewer 1 Report
Comments and Suggestions for Authors
- What was the objective of the study described in the paragraph?
- What was the study design and location of the study?
- What was the inclusion criteria for the patients included in the study?
- What was the primary endpoint of the study?
- How many patients were included in the full analysis set, and how were they divided into groups?
- Did the groups differ in clinical, demographic, or laboratory variables?
- What were the changes in LDL-C concentration in the statin monotherapy group and the combination therapy group?
- What percentage of patients in each group achieved the target LDL-C level?
- What are the conclusions of the study regarding the effectiveness of ezetimibe in combination with statins compared to statin monotherapy?
- By how much does the addition of ezetimibe to statin therapy increase the probability of achieving LDL-C targets, according to the study?
Reviewer 2 Report
Comments and Suggestions for Authors
This study aimed to investigate the efficacy of ezetimibe in lowering LDL-C levels using real-world data.
Major
1. The study setting addressed the institutions that participated in this study. Please show the inclusion criteria for the institution and the flowchart of enrollment. Wasn’t any intuition excluded? Were the institution and participants completely the same as those of the UNISON study?
Focusing reference #5, it showed “DA VINCI study,” is it the same as UNISON study?
2. Has the results of the UNISON study been published? If so, please cite it?
3. Real-world data generally consist of missing data, to some extent. How did the author treat the missing data? The proportion of missing data should be considered when discussing real-world scenarios.
4. How did the author determine the definition of achievement (Table 1) If there was scientific evidence for the achievement of LDL-C lowering, please address this evidence.
5. Baseline LDL-C in “plus ezetimibe” was greater than “monotherapy.” This is a critical problem because the efficiency of the LDL-C lowering effect largely depends on the baseline LDL-C level.
6. The most critical point of this study is the greater LDL-C-lowering effect derived from adding ezetimibe to statins. Therefore, scientific novelty is required.
Comments on the Quality of English LanguageThe English quality reaches some level to be published. More fluent English can be blush up.
Reviewer 3 Report
Comments and Suggestions for Authors
This was an important analysis of utilizing statins alone and in combination with ezetimibe to manage dyslipidemia. It is interesting that the combination therapy was more effective in achieving lipid goals. The results support the use of combination therapy. Heart disease affects a large proportion of the population throughout the world and using drugs to improve lipid profiles can benefit many individuals.
One note (and this may be a language issue) is that clarification needs to be made regarding, the changes observed (to ensure they are reported correctly) (e.g., percent change for each group and that the 29% difference is not between groups rather it is within one group).
Line 28-29:
Clarify please “The addition of ezetimibe 28 to statin therapy increases the probability of achieving LDL-C targets by 29%”
Line 59:
Perhaps define dyslipidemia (clinical values)
Line 62:
What was the age range? Patients aged ≥18 years to?
Line 66-67:
Possibly explain the difference between primary or secondary prevention of CVD
Line 124 - Table 1:
What determined the CVR (extreme, very high risk, high risk, moderate, low)?
Line 166 - Figure 1:
The differences between the medium gray color is very slight and hard to differentiate (Pitavastatin and Rosuvastatin in combination with ezetimibe (fixed dose combination)). Perhaps change to a different color or add the % to the name of the drug so the connection can be made.
Line 175-177:
This needs to be clarified “Thus, the addition of ezetimibe statin therapy increases the probability of achieving LDL-C targets by 29% (odds ratio 176 (OR) 0.7139 (0.4998-1.0091), p = 0.0493)”. In that the addition increased the goal LDL-C in the statin/ezetimibe group to 29% in comparison to 22% in the statin only group. It seems based on the statement that it was 29% more than the statin only group.
Comments on the Quality of English Language
A slight modification (specifically in reporting the percent change within groups and between groups).
Reviewer 4 Report
Comments and Suggestions for Authors
This study compared the efficacy and safety of statin monotherapy and statin plus ezetimibe combination in the real-world setting.
The strength of this manuscript is the sample size (n>250) is good and conclusion is logical.
However, there are weaknesses in the manuscript: First, the introduction is too simple. More background of statin and ezetimibe clinical studies are needed. Second, the description of results is hard to understand. Third, is there something wrong in table 1?
There are other minor revisions may need:
1. Line 35: Russia seems extra.
2. Line 92, 94: Keep the Visit 1 and Visit-1 in same.
3. Line 109: “medical history data family history of CVD” not understandable
4. The table in the Results should be table 2 not table 1
5. Keep the first column (Parameters) of table 2 consistent: n (%). Include the same problem in the bottom line of table 2 (Diuretics).
6. Figure 3 is not understandable.
Comments on the Quality of English LanguageModerate editing of English language required